# Regional Differences in the Diets of Adélie and Emperor Penguins in the Ross Sea, Antarctica

**DOI:** 10.3390/ani11092681

**Published:** 2021-09-13

**Authors:** Seo-Yeon Hong, Jong-Ku Gal, Bo-Yeon Lee, Wu-Ju Son, Jin-Woo Jung, Hyung-Sul La, Kyung-Hoon Shin, Jeong-Hoon Kim, Sun-Yong Ha

**Affiliations:** 1Division of Ocean Sciences, Korea Polar Research Institute (KOPRI), Incheon 21990, Korea; 96seoyeon@naver.com (S.-Y.H.); jkgal@kopri.re.kr (J.-K.G.); boyeon@kopri.re.kr (B.-Y.L.); swj5753@kopri.re.kr (W.-J.S.); hsla@kopri.re.kr (H.-S.L.); 2Department of Marine Sciences and Convergence Technology, Hanyang University, Ansan 15588, Korea; shinkh@hanyang.ac.kr; 3Department of Polar Sciences, University of Science and Technology, Daejeon 34113, Korea; 4Research Center for Endangered Species, National Institute of Ecology, Yeongyang 36531, Korea; jinwoojung81@gmail.com; 5Division of Life Sciences, Korea Polar Research Institute (KOPRI), Incheon 21990, Korea

**Keywords:** stable isotope analysis, Adélie Penguin, Emperor Penguin, Ross Sea, SIAR

## Abstract

**Simple Summary:**

Stable isotope analysis (SIA) and Stable isotope analysis in R (SIAR) model were used to identify the diet composition and regional differences of Adélie and Emperor penguins in Ross Sea region. Adélie Penguin at Cape Hallett fed on Antarctic krill and Adélie Penguin at Inexpressible Island fed on ice krill and Antarctic silverfish. Emperor Penguins fed on Antarctic silverfish regardless breeding site. Therefore, Adélie Penguin showed regional difference in the diet and Emperor Penguin showed no regional differences in the diet. These diet composition of penguins is affected by competition and distribution of prey, it is important to study the diet of penguins in relation to the sympathetic food sources needed to understand the changes in energy flows and Ross Sea ecosystems due to climate change.

**Abstract:**

To identify the dietary composition and characteristics of both Adélie (*Pygoscelis adeliae*) and Emperor (*Aptenodytes forsteri*) penguins at four breeding sites, we performed stable carbon (δ^13^C) and nitrogen (δ^15^N) isotope analysis of down samples taken from penguin chicks. Adélie Penguin chicks at Cape Hallett mostly fed on Antarctic krill (*Euphausia superba*; 65.5 ± 3.5%), a reflection of the prevalence of that species near Cape Hallett, and no significant differences were noted between 2017 and 2018. However, Adélie Penguin chicks at Inexpressible Island, located near Terra Nova Bay, fed on both Antarctic silverfish (*Pleuragramma antarctica*; 42.5%) and ice krill (*Euphausia crystallorophias*; 47%), reflecting the high biomass observed in Terra Nova Bay. Meanwhile, no significant difference was noted between the two breeding sites of the Emperor Penguin. Emperor Penguin chicks predominantly fed on Antarctic silverfish (74.5 ± 2.1%) at both breeding sites (Cape Washington and Coulman Island), suggesting that diet preference represents the main factor influencing Emperor Penguin foraging. In contrast, the diet of the Adélie Penguin reflects presumed regional differences in prey prevalence, as inferred from available survey data.

## 1. Introduction

Extreme environments such as Antarctica are sensitive to climate change, and sea-ice dynamics and temperature fluctuations affect organisms in the Antarctic environment [1,2]. Climate change affects the ecology of predators that feed on plankton, krill, and fish, representing an important part of the Antarctic food web that is highly related to the sea ice habitat [3,4,5,6]. The Ross Sea, which is a large marine protected area (MPA) in Antarctica, has been recognized for its ecological importance as a breeding site of the Adélie Penguin, Emperor Penguin, and Antarctic silverfish. In particular, it hosts breeding sites (e.g., Cape Crozier, with approximately 153,000 breeding pairs [7]) for over 100,000 pairs of Adélie Penguins (*Pygoscelis adeliae*), and more than 20% of the worldwide population of Emperor Penguins (*Aptenodytes forsteri*) exist in the Ross Sea region. Adélie and Emperor penguins are important predators in Antarctica, on feeding mid-trophic-level prey such as krill. In particular, the population of the Adélie Penguin, which is known as a krill-dependent species, is closely associated with krill biomass and serves as an indicator of climate change [8]. The Emperor Penguin is also sensitive to the Ross Sea sea-ice dynamics that are related to their prey habitat [9,10,11]. Thus, penguin feeding activity is strongly affected by changes in prey availability, so it is important to study penguin diets to understand Ross Sea food web dynamics. 

Krill, which is an important source of food in the Antarctic food web, is the species that accounts for the mid-trophic-level biomass in the Southern Ocean and provides an important link from primary producers to top predators [4,5,12,13]. Antarctic krill (*Euphausia superba*) and ice krill (*Euphausia crystallorophias*) comprise most of the krill biomass and represent a major food source for the top predators in Antarctica, such as penguins [3,6,14]. In the Ross Sea, Antarctic krill inhabit the continental slope and shelf, and are observed in large biomass and abundance in open water regions, whereas ice krill inhabit ice-covered and shallow coastal regions that are more related to sea ice [13,14]. Here, ice krill exhibit reduced biomass compared with Antarctic krill, but this abundance is increased closer to the coastal region [14]. Along with Antarctic krill and ice krill, Antarctic silverfish (*Pleuragramma antarctica*) is another important food source for an upper-trophic-level organism in the Ross Sea region. As an ice-obligate species, Antarctic silverfish reside in ice-covered zones, similarly to ice krill [13,15,16]. Considering these characteristics, it is also important to investigate the temporal and spatial distribution of prey species such as Antarctic krill, ice krill, and Antarctic silverfish in dietary studies of penguins of the Ross Sea region [17,18,19,20].

To accurately characterize the food web structure, various methods such as field observations, gut content analysis, and stable isotope analysis (SIA) are employed [21,22,23]. Gut content analysis offers intuitive food information but only provides a snapshot, and in the case of easily digested food, the amount can be underestimated [24]. In contrast, SIA provides relatively long-term diet information and diminishes the errors associated with short-term analysis [25]. Therefore, analysis of stable carbon and nitrogen isotopes in organisms is an efficient method to assess feeding relationships, energy flow, and trophic position in food webs [26,27,28]. However, to the best of our knowledge, SIA data of Adélie and Emperor penguins in the Ross Sea have seldom been reported, with the exception of the study of Pilcher et al. [29], who reported on the mercury concentration and carbon and nitrogen stable isotope values of the Adélie and the Emperor penguin in the western Ross Sea region. However, the authors focused only on the accumulation of mercury according to trophic level based on the analysis of stable nitrogen isotopes, and did not include further discussion of the feeding relationship between penguins and energy flow in the Ross Sea. The objectives of this study were (1) to estimate the diet contribution and preference for Adélie and Emperor penguins, and (2) to identify the latitudinal effect on diet composition of penguins at given breeding sites.

## 2. Materials and Methods

### 2.1. Study Area and Sample Collection

Antarctic krill and ice krill samples were collected from 31 October 2018 to 16 April 2019 during the R/V ARAON Expedition (ANA09B) in the western Ross Sea (Figure 1). Antarctic krill samples were collected using a frame net (330 µm mesh) on 19–20 January 2019. Ice krill samples were collected using a bongo net (505 µm mesh) and rectangular net (330 µm mesh) on 16–17 January 2019. The collected krill samples were immediately classified onboard based on microscopic analysis. Sorted samples were stored at −20 °C.

Chick carcasses of Adélie and Emperor penguins were collected at Cape Hallett, Inexpressible Island, Cape Washington, and Coulman Island. All chick samples were collected by hand, and only undamaged whole bodies were collected. The growing feathers of chicks contain diet components accumulated during the breeding season, which are a good representation of the penguins’ diet specifically during the breeding season. They can also reflect the foraging activities of adult penguins during the breeding season, since before molting, the chicks are dependent on the adult diet for food. In contrast, the adult feathers are expected to reflect feeding information for the entire molt period. Thereore, we conducted our study by analyzing the carbon and nitrogen isotopic signatures of the growing feathers (also known as down) of chicks. According to Vasil et al. [31], there are no significant differences in the isotope composition of living and dead penguin chick tissue (feather, down, and toenail), so possible effects of physiological stress and starvation on isotope signature were discounted. All samples were frozen at −20 °C after collection.

### 2.2. Sample Preparation and Stable Isotope Analysis

Whole body samples of Antarctic krill and ice krill were freeze-dried and homogenized using a mortar and pestle. To remove inorganic carbon, samples were acidified in 1 M hydrochloride by shaking for 12 h. The acid was decanted after centrifugation, and the remaining samples rinsed with distilled water three times. The samples with inorganic carbon removed were defatted three times using a 2:1 (*v*/*v*) chloroform/methanol mixture. Pretreated samples were used for stable carbon isotope analysis. Untreated samples were used for nitrogen stable isotope analysis to avoid variations in nitrogen isotope values due to hydrochloride treatment [32].

Penguin breast down samples used for SIA were rinsed using distilled water followed by a 2:1 (*v*/*v*) chloroform/methanol mixture. Rinsed samples were air-dried and cut into small fragments using stainless steel scissors. For SIA, all samples were weighed and enveloped using tin capsules.

The δ^13^C and δ^15^N ratios were determined using an isotope ratio mass spectrometer (Isoprime Vision IRMS, Germany) coupled with an elemental analyzer (Elementar select EA, Germany). Stable isotope ratios are expressed in δ notation in per mil units (‰) according to the following equation:δX (‰) = (R_sample_/R_standard_ − 1) × 1000
where X is ^13^C or ^15^N and R_sample_ is the corresponding ratio ^13^C/^12^C or ^15^N/^14^N. Vienna Pee Dee Belemnite (VPDB) and air were used as standards for δ^13^C and δ^15^N, respectively. CH3 (−24.7‰) and N–1 (0.4‰) (International Atomic Energy Agency, IAEA) were used to run standards for δ^13^C of and δ^15^N, respectively. These standards were each analyzed in all 12 samples. The analytical deviations were <0.2‰ for the SIA results of both carbon and nitrogen.

### 2.3. Statistical Analysis

The *t*-test was performed using the statistical package in R to determine if there were significant differences between regions. Prior to the *t*-test, we used a Shapiro–Wilk test to determine δ^13^C and δ^15^N data normality. We performed a paired-sample *t*-test and two-sample *t*-test. The paired-sample *t*-test was performed to compare annual differences in Cape Hallett (2017/18 versus 2018/19); the two-sample *t*-test was performed to compare between Adélie Penguin breeding sites (2017/18 Cape Hallett versus 2017/18 Inexpressible Island) and Emperor Penguin breeding sites (2018/19 Cape Washington versus 2018/19 Coulman Island).

The relative contributions of potential food sources to the diets of penguins were estimated using stable isotope analysis in R (SIAR; [33]). For SIAR modeling, we used previously reported stable isotope ratios of Antarctic silverfish ([30]; Table 1) and the diet to feather trophic enrichment factor (TEF). The TEF for penguin was 1.3 ± 0.4‰ for carbon and 3.5 ± 0.5‰ for nitrogen, which were based on TEF values reported for the Gentoo Penguin [34]. Confidence intervals were set at 95, 75, and 25%, and the SIAR model was applied to penguins and their diets.

## 3. Results

### 3.1. Isotopic Signatures of Penguin Chicks and Their Prey

The δ^13^C and δ^15^N levels in Antarctic krill were found to be lower than those of ice krill, reflecting the difference in main food sources (pelagic phytoplankton sources versus sympagic sources [35]). Isotopic values were significantly different between ice krill and Antarctic krill (*p* < 0.0001; *t*-test; Table 1; Figure 2).

The isotopic signatures of Adélie Penguin chicks in Cape Hallett exhibited no significant difference regardless of year (*p* = 0.1674 in δ^13^C, *p* = 0.3794 in δ^15^N; *t*-test), but those in Cape Hallett and Inexpressible Island exhibited significant differences in both δ^13^C and δ^15^N values (*p* < 0.0001; *t*-test). The mean carbon stable isotope values of Emperor Penguin chicks showed significant difference (*p* = 0.008687; *t*-test), but nitrogen isotopic signatures were not significantly different between the two breeding sites (*p* = 0.3466; *t*-test). The carbon and nitrogen stable isotope values of Emperor Penguin chicks were more enriched than those observed for Adélie Penguin chicks (Table 1; Figure 2). This shows that krill and Antarctic silverfish, the mid-trophic-level species, are linked to algae as the primary producers and to top predator penguins.

### 3.2. Diet Proportions of Penguin Chicks

The diets of Adélie Penguin chicks at Cape Hallett showed higher proportions of krill compared with the other groups (Table 2; Figure 3). In particular, the proportion of Antarctic krill was the highest for Adélie chicks at Cape Hallett at 68 and 63% in 2017/18 and 2018/19, respectively. The diet composition of Adélie Penguin chicks on Inexpressible Island differed from those at Cape Hallett, with differences in proportions of Antarctic krill and Antarctic silverfish (11 and 43%, respectively). Antarctic silverfish represented the largest component of the diets of Emperor Penguin chicks at 76 and 73% in Cape Washington and Coulman Island, respectively.

## 4. Discussion

### 4.1. Influence of Latitudinal Diet Differences for Adélie Penguin

Antarctic krill, ice krill, and Antarctic silverfish represent the major prey species of upper predators, including penguins in the Ross Sea [13,16,19,36,37]. The diets of Adélie Penguin chicks at Cape Hallett exhibited a high proportion of Antarctic krill at 65.5 ± 3.5% (the combined Antarctic krill and ice krill proportion were more than 95%; Table 2) and no significant difference was noted in yearly comparisons. The diets of Adélie Penguin chicks on Inexpressible Island exhibited a high proportion of Antarctic silverfish compared with those at Cape Hallett (Table 2), and a significant difference was noted for Adélie Penguins depending on whether they were from Cape Hallett or Inexpressible Island. Such a difference in diet could exist due to prey availability depending on latitudinal location [19,37]. Cape Hallett and Inexpressible Island are located approximately 330 km apart (Figure 1). Therefore, the differences in diet composition could reflect the regional characteristics of prey distribution. Antarctic krill are important prey for Adélie Penguins given their large biomass at the continental shelf and slope in the northern Ross Sea [13,14]. In contrast, ice krill and Antarctic silverfish exhibit relatively lower biomass than Antarctic krill throughout the western Ross Sea, but their biomass and abundance are high near the coast, especially in Terra Nova Bay [13,14]. Consequently, Adélie Penguins at Inexpressible Island are more likely than those at Cape Hallett to encounter ice krill and Antarctic silverfish.

It should be noted that errors may have occurred in estimating the food contribution of Adélie Penguins in the 2017/18 austral summer because the data of potential food sources (krill and Antarctic silverfish) were obtained for a different year. Antarctic and ice krill data acquired during the 2018/19 austral summer season, and Antarctic silverfish data were obtained from Pinkerton et al. (2013). However, isotopic values of Antarctic silverfish in the Ross Sea do not show annual and spatial variations during austral summer (cf. [38,39]). Moreover, there was clear isotope discrimination between ice krill and Antarctic krill, and the data for isotope distribution of krill in this study did not show a significant difference from the previously reported data [24]. Furthermore, based on the fact that the habitat distribution of the krill also showed a difference, it can be inferred that the isotope ratios of potential food sources do not change significantly during the austral summer period. Therefore, our estimation of feeding contribution is reasonable.

Competition affects the diet composition of penguins. Indeed, intraspecific competition of Adélie Penguins can occur within Cape Hallett or with adjacent larger colonies, such as Cape Adare and Possession Island (combined 338,000 breeding pairs [7]). According to Lyver et al. [19], Adélie Penguins from Cape Hallett (19,744 breeding pairs [19]) exhibit relatively lower intraspecific competition. Nevertheless, intraspecific competition can occur between penguins from Cape Hallett and adjacent breeding sites, given the numerous breeding pairs of Adélie Penguins close to Cape Hallett (e.g., Cape Adare and Possession Island; combined 338,000 breeding pairs [40]). Thus, the Adélie Penguins could have adjusted their behavior by foraging farther and for longer to reduce competition [19]. Antarctic krill are present at high levels in the northern region (including Cape Hallett) of the Ross Sea region [4,13,14,16], so Adélie Penguins at Cape Hallett which had a wider foraging area would have been found it easier to consume Antarctic krill. As shown in Figure 3, the diet of Adélie Penguins at Cape Hallett exhibited a high Antarctic krill contribution regardless of year, whereas ice krill and Antarctic silverfish occupied a relatively small proportion (Table 2).

In contrast, we excluded intraspecific competition at Inexpressible Island because the number of breeding pairs is small (~24,000 breeding pairs [7,19]) and no larger breeding sites are located near Inexpressible Island. The diets of Adélie Penguin at Inexpressible Island exhibited a relatively high proportion of Antarctic silverfish compared with those at Cape Hallett (Table 2), based on the enrichment of the associated carbon and nitrogen isotopes of samples from Inexpressible Island (Figure 2). Antarctic silverfish comprise most of the fish biomass in the Ross Sea region, and high biomass levels of Antarctic silverfish larva and juveniles are observed under the Terra Nova Bay sea ice within a 200 m depth, which corresponds with the foraging depth of Adélie Penguins. Thus, penguins at Inexpressible Island may have been able to forage Antarctic silverfish more easily. The proportion of ice krill was increased compared with that of Antarctic krill in the diets of Adélie Penguins. At Terra Nova Bay, ice krill and Antarctic silverfish exhibit high abundance, whereas Antarctic krill are present at a relatively low abundance [41]. Thus, it appears that Adélie Penguins at Inexpressible Island had less opportunities to encounter Antarctic krill [13,14]. Consequently, Adélie Penguins at Inexpressible Island consumed a relatively higher abundance of ice krill and Antarctic silverfish in their foraging area, resulting in a significant difference in the estimated diet composition and isotopic signatures compared with penguins at Cape Hallett (Figure 3, Table 1). In addition, Adélie Penguins feed more fish to chicks during the breeding season because fish are high in calories and easily digested, representing an effective food source for chick rearing [10,18,42]. However, Ainley et al. [41] suggests that Adélie Penguins are affected by latitudinal location (regional difference) in the Ross Sea region, given the increased biomass of Antarctic krill. Thus, penguins at Cape Hallett located the northern Ross Sea feed on more Antarctic krill during the breeding season. Meanwhile, the diet of another colony on Ross Island located in the southern Ross Sea is almost completely composed of fish (mainly Antarctic silverfish) and ice krill [19,41,43].

The Adélie Penguin is known as an Antarctic krill-dependent species [19,20,37]. However, the results of this study suggest that the Adélie Penguin is not always the dominant species consuming Antarctic krill and may be influenced by latitudinal diet differences. Adélie Penguins at Cape Hallett fed primarily on Antarctic krill, and those at Inexpressible Island foraged ice krill and Antarctic silverfish [44,45], which were more abundant regionally compared to Cape Hallett.

### 4.2. Diet Preference of Emperor Penguin

Emperor Penguins at Coulman Island in the northern region consumed more Antarctic krill compared to those at Cape Washington, but had a lower proportion of ice krill in their diets (Table 2). Antarctic silverfish accounted for a high and similar proportion of the diets of Penguins at both breeding sites considering the estimated diet contribution (Figure 3, Table 2). This suggests that Antarctic silverfish play an important role in the Emperor Penguin diet. Carbon isotopic signatures of Emperor Penguin at Cape Washington were slightly higher, showing a significant difference in δ^13^C. On the other hand, δ^15^N exhibited no significant difference, suggesting no trophic segregation of Emperor Penguins between breeding sites. The discrepancy in δ^13^C values of Emperor Penguins seemed to be due to the slightly low proportion of Antarctic krill in the diet of penguins at Cape Washington (Table 2), in contrast to the high biomass of Antarctic krill near Coulman Island and the high biomass of ice krill and Antarctic silverfish at Cape Washington near Terra Nova Bay [13,14]. Antarctic silverfish were observed throughout the Ross Sea and, in particular, very high juvenile biomass was observed near Terra Nova Bay [13,14] because they hatch and grow under the sea ice there, which remains until mid-January [15]. The Emperor Penguins of Coulman Island showed a high diet proportion of Antarctic silverfish, which could have been due to availability given the presence of fast ice. However, a high biomass of Antarctic krill is also observed near Coulman Island. Regarding the prey distribution near Coulman Island, there are high proportions of Antarctic krill and Antarctic silverfish. The Coulman Island Emperor Penguin diet showed a high proportion of Antarctic silverfish and a low proportion of Antarctic krill. This suggests that Emperor Penguins prefer silverfish to krill. In addition, Emperor Penguins, which can forage farther than Adélie Penguins, would be less affected by the inshore prey distribution, which is generally variable closer to the coast. The foraging areas of Adélie and Emperor penguins overlap between Cape Hallett and Coulman Island, as well as between Inexpressible Island and Cape Washington, given the closeness of these locations. However, we assumed that potential competition between Adélie and Emperor penguins is low given their different foraging behaviors, diving depth, and foraging distances.

Weddell seals (*Leptonychotes weddellii*) in the Ross Sea exhibit similar diet (Antarctic silverfish are dominant prey) and foraging behaviors (especially juvenile Weddell seals) to Emperor Penguins [10,46,47]. The austral summer season, which includes November, is the rearing period of Emperor Penguin chicks and Weddell seal pups. In the pup rearing period, Weddell seal pups are still nursed by their mothers and adult Weddell seals dive deeper than adult Emperor Penguins. Although sizes of Antarctic silverfish consumed by Emperor Penguins and Weddell seals are nearly identical, the adult Weddell seals had more enhance diving behavior than Emperor Penguins and occupied the benthic sources not found in Emperor Penguins [48]. Hence, potential interspecific competition between the two predator species may be low during the sampling season in this study due to the segregation of foraging depth between these two predator [47]. Thus, no significant differences in the diets of Emperor Penguin are noted based on region, suggesting a minimal influence of prey abundance compared to the Adélie Penguin and competition with other animals, given that the fish preference of Emperor Penguins is an important factor influencing foraging strategies. This study had limitation of the small sample size (*n* = 5) due to the specificity of the sampling area. The field environmental could not be predicted, so we have to make more effort for sampling in further studies. Nevertheless, diet composition reflects the changing climate as well as current ecological environment, and it will be helpful in the study of anthropogenic pollutions such as trace metal and oils [29,49,50], so it is important to identify the diet composition by considering many factors such as dietary distribution, competition, and foraging behavior.

## 5. Conclusions

In this study, we characterized the diets of both Adélie and Emperor penguins, which are one of the key predators in the Antarctic ecosystem. The estimated diet composition of Adélie Penguin exhibited significant differences between Cape Hallett and Inexpressible Island populations, and regional prey distribution may have affected the availability of species (Antarctic krill, ice krill and Antarctic silverfish) in the diet of Adélie Penguins. The diets of Emperor Penguins exhibited no significant differences based on breeding sites, so prey preference represents a more important factor than prey abundance in Emperor Penguin. The present study supports the importance of sympagic food sources such as ice krill and Antarctic silverfish. Therefore, changes in the food web dynamics and energy flows in the Antarctic ecosystem might arise from changes in sympagic food sources due to climate change.

## Figures and Tables

**Figure 1 animals-11-02681-f001:**
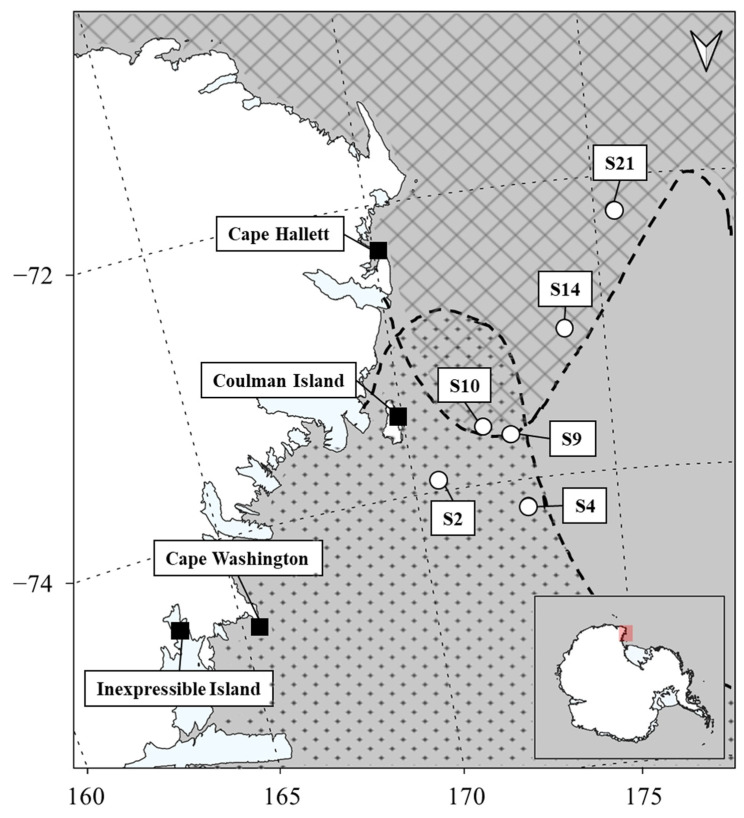
Study area. Penguin colonies (square), krill sampling area (white circle), distribution of Antarctic krill (cross line), and distribution of ice krill (dot pattern) in the western Ross Sea, Antarctica. Distribution of krill modified from the Ross Sea Region Marine Protected Area Research and Monitoring Plan, CCAMLR. Details are presented in Table 1.

**Figure 2 animals-11-02681-f002:**
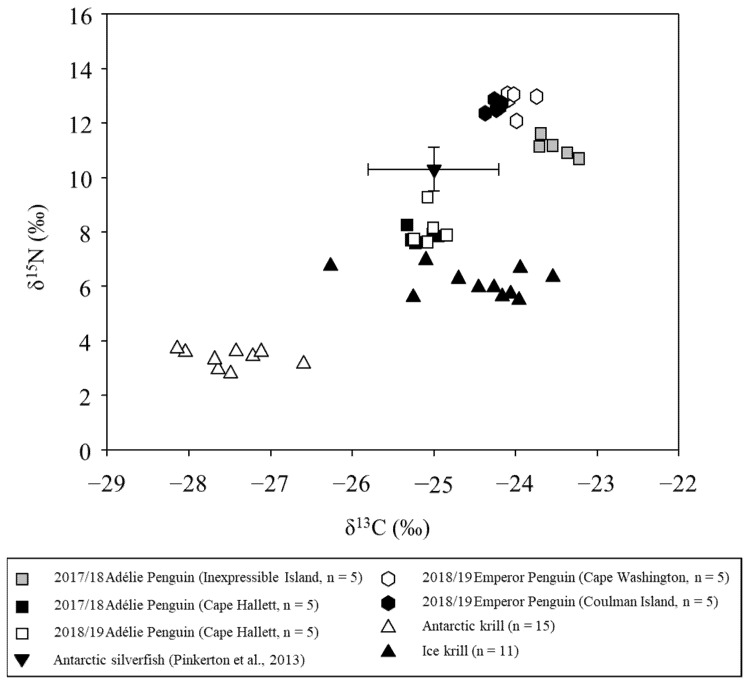
Stable isotopic signatures of penguin chick down and their main prey. Black square: Adélie Penguin chicks at Cape Hallett in 2017; white square: Adélie Penguin chicks at Cape Hallett in 2018; gray square: Adélie Penguin chicks at Inexpressible Island in 2018 (duration of 2017 breeding season); white hexagon: Emperor Penguin chicks at Cape Washington in 2018; black hexagon: Emperor Penguin chicks at Coulman Island in 2018; downward-pointing black triangle: Antarctic silverfish (*P.*
*antarctica*); upward-pointing black triangle: ice krill (*E. crystallorophias*); and upward-pointing white triangle: Antarctic krill (*E. superba*).

**Figure 3 animals-11-02681-f003:**
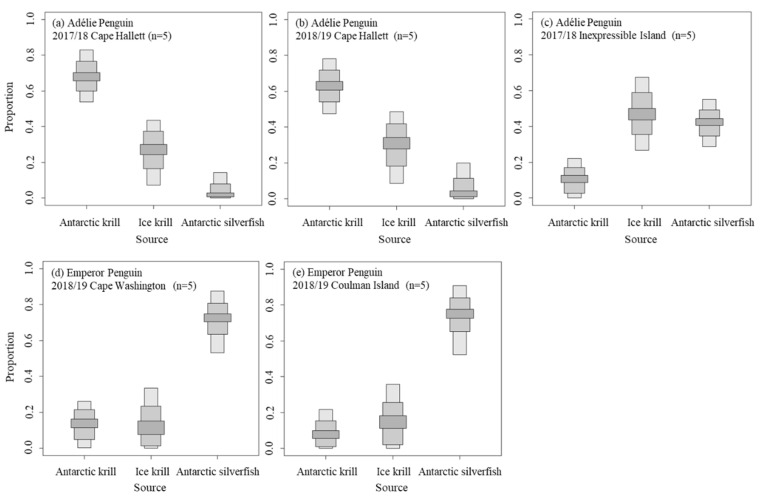
Proportion of main prey (Antarctic krill, ice krill, and Antarctic silverfish (%)) in the diets of Adélie Penguin chicks (**a**–**c**) and Emperor Penguin chicks (**d**,**e**). The gray bars of different widths from dark to light indicate the 95, 75, and 25% confidence intervals.

**Table 1 animals-11-02681-t001:** Sampling details and stable isotope signatures of prey items and penguin chicks.

Common Name	N	Analyzed Tissue	δ^13^C (‰)	δ^15^N (‰)	Sampling Date (yy/mm)	Sampling Area	References
Ice krill	11	Whole body	−24.52 ± 0.77	6.12 ± 0.51	19/01	S2, S4, S9	This study
Antarctic krill	15	Whole body	−27.41 ± 0.57	3.65 ± 0.58	19/01	S10, S14, S21	This study
Adélie Penguin	5	Down	−25.16 ± 0.17	7.86 ± 0.25	17/12 (guard–créche stage)	Cape Hallett	This study
	5	Down	−25.05 ± 0.14	8.13 ± 0.67	18/12 (guard–créche stage)	Cape Hallett	This study
	5	Down	−23.50 ± 0.21	11.10 ± 0.34	18/01 (créche stage)	Inexpressible Island	This study
Emperor Penguin	5	Down	−23.99 ± 0.15	12.81 ± 0.42	18/11 (créche stage)	Cape Washington	This study
	5	Down	−24.24 ± 0.08	12.60 ± 0.20	18/11 (créche stage)	Coulman Island	This study
Antarctic silverfish	140	Whole body	−25.00 ± 0.80	10.30 ± 0.80	08/02–03	Western Ross Sea	[30]

**Table 2 animals-11-02681-t002:** Diet composition of Adélie and Emperor penguin chicks based on the SIAR model. Values are presented as the mean estimates with 95% credibility interval.

Year	Species	Breeding Site	Diet Proportion (%)		
			Antarctic Krill	Ice Krill	Antarctic Silverfish
2017/18	Adélie Penguin	Cape Hallett	68 (66–70)	27 (24–30)	2 (0.7–2.8)
2018/19	Adélie Penguin	Cape Hallett	63 (60–65)	31 (28–34)	3 (1.2–4.5)
2017/18	Adélie Penguin	Inexpressible Island	11 (8.6–13)	47 (44–50)	43 (41–44)
2018/19	Emperor Penguin	Cape Washington	8 (5.6–9.9)	15 (11–18)	76 (73–78)
2018/19	Emperor Penguin	Coulman Island	14 (11–16)	11 (7.6–15)	73 (70–75)

## Data Availability

The data presented in this study are available on request from the corresponding author.

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
