# Peer review of "Regional Differences in the Diets of Adélie and Emperor Penguins in the Ross Sea, Antarctica"

_animals, 2021, doi:10.3390/ani11092681_

Round 1

Reviewer 1 Report

The reviewers comments and suggestions for authors are in the attached file.

Author Response

We appreciated for your comments. All comments were revised in manuscript and we responded the line number and revised text, please check the below.

1. The common (English) names of species should be capitalized according to the World list of English bird names - check the entire text. It is corrected in the text. (emperor penguin  Emperor penguin)

2. Page 1, Introduction, lines 4 and 5 - the Latin names of the Еmperors and Adélie penguins at the first mention in the body of the article should not have abbreviations - It is corrected in the text.

3. In the Introduction, combine the last two paragraphs into one paragraph. It is corrected in the text.

4. Suggestions for improving Figure 1. It is corrected in the text.

А. In the upper left corner, you need an inset (a small map), where the Ross Sea Region is shown with an indication of the place (point) of the research. It is advisable to place a compass. Then make explanations in the legend to Figure 1. 

В. Notation S2-S21 (Sampling Areas) - change the black color to white, since black is lost on a gray background. 

C. In the legend for Figure 1. 
replace "open circle" with "white circle". D. In the legend for 
Figure 1. fix CCAMILR to CCAMLR. 

E. Common names of species in the legends to Figures 1-3 do not have Latin names everywhere – fix the discrepancy.5. 
2.1. Study Area and Sample Collection. «All procedures, including those involving animals, were carried out in accordance with the ethical standards of the institutional guidelines the Animal Welfare Ethical Committee and the Animal Experimental Ethics Committee of the Korea Polar Research Institute (KOPRI, South Korea)» - please delete the sentence in this section. This paragraph is a duplicate and is located in the Institutional Review Board Statement. It is corrected in the text.

6. Page 6, the 6th and 11th lines from the bottom – "combined 338,000" (insert breeding pairs, individuals?). 7. Page 7, 4th line from the top – (24,000 breeding pairs-insert). (line 221, 226 and 235, “breeding pairs”)

8. In the Discussion section in the last paragraph it is useful to indicate (1-2 sentences) why the Weddell seal in the study area of the Ross Sea Region can be considered an important competitor to the Еmperor Рenguin. 

(line 309-310. “Also, they are one of the key predator with penguins.”)

9. After the first sentence of the last paragraph in the Discussion section (Weddell seals (Leptonychotes weddellii) in the Ross Sea exhibit very similar diet (Antarctic silverfish are dominant prey) and foraging behaviors (especially juvenile Weddell seals) to emperor penguins), references are needed. 

10. At the end of the Discussion section, it is useful to indicate the prospects for the development of the topic (2-3 sentences). (line 323-329. “This study had limitation of the small sample size (n=5) due to the specificity of the sampling area. The field environmental could not be predicted, so we have to make more effort for sampling in further studies. Nevertheless, diet composition reflects the changing climate as well as current ecological environment, and it will be helpful in the study of anthropogenic pollutions such as trace metal and oils, so it is important to identify the diet composition by considering many factors such as dietary distribution, competition, and foraging behavior.)

11. Section Conclusion, first sentence: In this study, we characterize the diets of both Adélie and emperor penguins, which are key predators in the Antarctic ecosystem. But whales, for example, are also key predators of Antarctic ecosystems. Therefore, it is more correct to say that Adélie and Emperor Penguins are one of the key predators. Make changes. It is corrected in the text.

12. Conclusion section, second sentence from the top: The estimated diet composition of Adélie penguin exhibited significant differences between Cape Hallett and Inexpressible Island populations, and regional prey distribution may have affected the availability of species (what species? list) in the diet of Adélie penguins. It is corrected in the text. (line 332. “Antarctic krill, ice krill and Antarctic silverfish).

Reviewer 2 Report

The authors present a nice study on variation in feeding patterns in two species of penguins in the Ross Sea of Antarctica.  The study is scientifically sound, well written, and worthy of publication.  Only very minor suggestions, as follows:

(please note i have suggested to the editors that adding line numbers would GREATLY help in this process)

Page 2: , first full paragraph starting "Krill, which.."  the 5th line, remove "a":  "...For the top predators in Antarctica, such as penguins..." 

Figure 1:  I think it would be good to have a small map showing all of Antarctica, with then a "blow up" of the region of interest."  It gives a better context for geographic variation, etc.

Methods: I am not an expert in isotopic analysis, so will assume the metnods are correct. 

Results: Figure 3 is never referred to in the results.  It only appears later in the discussion.  IT might be useful to include it here, as it does a very nice job of showing the difference in 3 species, as well as differences between locations.

Pg 6 - Discussion, first paragraph starting "Antarctic krill, ice krill, and..."  6 line from bottom:  "Antarctic frill are important prey for Adelie PENGUINS, given their..." 

Pg 7 - Paragraph starting "In contrast, we excluded intraspecific...."  6th line from bottom:  "However, [43] suggests that Adelie PENGUINS ARE affected by latitudinal..." 

pg 9 - paragraph on Weddell seals.  I"m not sure this is very relevant.  Indeed, if it is to be included, there should be references relevant to the data/information about Weddell seals.   I"m not sure this paragraph adds much to the paper, or if it is felt to be important, then  more information needs to be discussed.

Author Response

#Reviewer 2

We thank the Reviewer 2 for valuable comments. All comments were revised in manuscript and we provided the line number and revised text below.

Page 2: , first full paragraph starting "Krill, which.."  the 5th line, remove "a":  "...For the top predators in Antarctica, such as penguins..." It is corrected in the text.

Figure 1:  I think it would be good to have a small map showing all of Antarctica, with then a "blow up" of the region of interest."  It gives a better context for geographic variation, etc. It is corrected in the text.

Methods: I am not an expert in isotopic analysis, so will assume the metnods are correct. It is corrected in the text.

Results: Figure 3 is never referred to in the results.  It only appears later in the discussion.  IT might be useful to include it here, as it does a very nice job of showing the difference in 3 species, as well as differences between locations. It is corrected in the text.

Pg 6 - Discussion, first paragraph starting "Antarctic krill, ice krill, and..."  6 line from bottom:  "Antarctic frill are important prey for Adelie PENGUINS, given their..."  It is corrected in the text.

Pg 7 - Paragraph starting "In contrast, we excluded intraspecific...."  6th line from bottom:  "However, [43] suggests that Adelie PENGUINS ARE affected by latitudinal..." It is corrected in the text.

pg 9 - paragraph on Weddell seals.  I"m not sure this is very relevant.  Indeed, if it is to be included, there should be references relevant to the data/information about Weddell seals.   I"m not sure this paragraph adds much to the paper, or if it is felt to be important, then  more information needs to be discussed. (line 314-316. “Although sizes of Antarctic silverfish consumed by Emperor penguins and Weddell seals are nearly identical, the adult Weddell seals had more enhance diving behavior than Emperor penguins and occupied the benthic sources not found in Emperor penguins”, and we added the some references.)

Round 2

Reviewer 1 Report

Please finde the comments in the attachment.

Author Response

We thank for your kindly comments. We corrected all comment in the manuscript.

  1. Correct spelling of names:Adélie Penguin, Adélie penguins - check the entire text and captions to the figures, for example, line 269. It is corrected in the entire text.
  2. Suggestions for improving Figure 1. The compass and the inset are very good! Notation S2-S21 (Sampling Areas) - they should be bright white (!), but not gray-white. For better visualization, the size of the symbols can be increased, then you can use the black color of the symbols. It is corrected in the text. For better visualization, we changed the notation S2-S21 using text box border.
  3. The first sentence of the last paragraph in the Discussion section (lines 331-333): Weddell seals (Leptonychotes weddellii) in the Ross Sea exhibit very similar diet (Antarctic silverfish are dominant prey) and foraging behaviors (especially juvenile Weddell seals) to emperor penguins and one of the key predators with penguins [10,37,46]. Delete the fragment «and one of the key predators with penguins». It is corrected in the text. (line 310)
